# Bilayer Forming Phospholipids as Targets for Cancer Therapy

**DOI:** 10.3390/ijms23095266

**Published:** 2022-05-09

**Authors:** Celine Stoica, Adilson Kleber Ferreira, Kayleigh Hannan, Marica Bakovic

**Affiliations:** 1Department of Human Health and Nutritional Science, College of Biological Sciences, University of Guelph, 50 Stone Road East, Guelph, ON N1G 2W1, Canada; cstoica@uoguelph.ca (C.S.); hannahk@uoguelph.ca (K.H.); 2Department of Immunology, Laboratory of Tumor Immunology, Institute of Biomedical Science, University of São Paulo, São Paulo 05508-000, Brazil; kleber@alchemypet.com.br; 3Department of Oncology, Alchemypet—Veterinary Dignostic Medicine, São Paulo 05024-000, Brazil

**Keywords:** phospholipids, cancer, fatty acids, metabolism, Kennedy pathway

## Abstract

Phospholipids represent a crucial component for the structure of cell membranes. Phosphatidylcholine and phosphatidylethanolamine are two phospholipids that comprise the majority of cell membranes. De novo biosynthesis of phosphatidylcholine and phosphatidylethanolamine occurs via the Kennedy pathway, and perturbations in the regulation of this pathway are linked to a variety of human diseases, including cancer. Altered phosphatidylcholine and phosphatidylethanolamine membrane content, phospholipid metabolite levels, and fatty acid profiles are frequently identified as hallmarks of cancer development and progression. This review summarizes the research on how phospholipid metabolism changes over oncogenic transformation, and how phospholipid profiling can differentiate between human cancer and healthy tissues, with a focus on colorectal cancer, breast cancer, and non-small cell lung cancer. The potential for phospholipids to serve as biomarkers for diagnostics, or as anticancer therapy targets, is also discussed.

## 1. Introduction

The phospholipids phosphatidylcholine (PC), phosphatidylethanolamine (PE), and phosphatidylserine (PS) are critical components of biological membranes, forming a fluid lipid bilayer that serves to maintain the structural integrity and selective permeability of cells. Phospholipids have numerous functions in cell division, autophagy, and apoptosis and serve as second messengers, protein chaperones, and receptors for the recruitment of membrane-bound proteins [1,2].

Choline and ethanolamine are nutrients obtained through diet [3,4,5]. The main fate of these nutrients is to enter the phospholipid Kennedy pathway, which is responsible for de novo synthesis of PC and PE (Figure 1) [1,2,3,4,5,6,7,8]. The PC and PE branches of the Kennedy pathway are known as the CDP:choline and the CDP:ethanolamine pathways, respectively. In these pathways, choline is converted to PC, and ethanolamine is converted to PE via a series of analogous enzymatic reactions. The third bilayer phospholipid PS is synthesized in the endoplasmic reticulum (ER) from PC and PE by ‘base-exchange’ reactions catalyzed by PS synthase 1 (PTDSS1) and PS synthase 2 (PTDSS2), respectively. In addition, PS is decarboxylated to PE by mitochondrial PS decarboxylase (PISD), and PE is methylated to PC by PE methyltransferase (PEMT) which helps in maintaining the membrane phospholipid ratio and homeostasis [1,2,3,4,5,6,7,8,9,10].

In the CDP:choline pathway, choline is first phosphorylated in an ATP-dependent reaction by choline kinase (CK) to produce phosphocholine [2]. Phosphocholine cytidylyltransferase (CCT/PCYT1) then uses phosphocholine and CTP to form CDP-choline [2]. Finally, this high-energy intermediate is coupled with DAG by choline phosphotransferase (CPT or CEPT) to produce PC [2,11].

The CDP:ethanolamine pathway occurs much the same way. Briefly, phosphoethanolamine is produced by ethanolamine kinase (ETNK), followed by conversion to CDP-ethanolamine by phosphoethanolamine cytidylyltransferase (ECT/PCYT2), and ethanolamine phosphotransferases (CEPT and EPT1/SELENO1) catalyze the final reaction, yielding PE [2,12]. In addition, phosphotransferases can use both CDP-choline and CDP-ethanolamine [13,14,15,16], but PCYT1 and PCYT2 are highly specific for their substrates and are the primary regulatory enzymes in their respective branches of the Kennedy pathway [2,17,18,19].

PS is synthesized from PC and PE by ‘base-exchange’ reactions using PTDSS1 and PTDSS2, respectively [6,7]. While PS is not as abundant as PC or PE, it is an essential inner membrane phospholipid that plays a critical role in mitochondrial function and apoptotic cell death [20,21].

Phospholipid regulation is important for proper physiological function, and disruptions in the Kennedy pathway have been linked to a variety of adverse health outcomes [2,3]. For instance, choline is well known for its importance in neurodevelopment and cognition [4]. Recently, we discovered a novel neurodegenerative disease CONATOC (neurodegeneration, childhood-onset, with ataxia, tremor, optic atrophy, and cognitive decline; OMIM # **#** 618868), characterized by a mutation in CTL1 choline transporter gene SLC44A1, resulting in reduced PC synthesis, although PC content remained unchanged due to excessive remodeling of PE and other glycerolipids [22]. Inadequate PC synthesis due to other disruptions in the CDP:choline pathway has also been linked to lipodystrophy, insulin resistance, diabetes, and liver damage [4,23].

Disequilibrium of the membrane PC to PE ratio can activate an unfolded protein response in the endoplasmic reticulum (ER) that can lead to apoptosis [3]. Phospholipid disequilibrium also contributes to obesity, hypertriglyceridemia, and insulin resistance [3,24]. The hearts of male mice deficient in *Pcyt2* have reduced PE synthesis, an altered PC:PE ratio, and modified fatty acid composition [24]. As a result, the male mice specifically developed cardiac dysfunction and hypertension [24].

Alterations in phospholipid metabolism have a profound effect on membrane structure, and consequently, its function [25]. Cancer cells require more membranes for rapid cell proliferation [26,27], and thus, cancer cells have increased phospholipid metabolism [26,28,29,30]. Altered fatty acid compositions have also been identified as characteristic of cancers [25,31,32]. Furthermore, increased de novo fatty acid synthesis is associated with aggressive cancers and a worse disease prognosis [26,30,33,34]. A link between metastatic disease pathogenesis and several enzymes involved in fatty acid synthesis has been described [34]. Under hypoxic conditions, cancer cells also increase exogenous fatty acid uptake [33]. Together, this dysregulation in fatty acid content and lipogenesis results in increased phospholipid saturation, rendering cancer cell membranes less susceptible to peroxidation-mediated and oxidative stress-mediated cell death [26,33]. Modifications in phospholipid levels can also alter crucial cellular signaling pathways, (e.g., cell proliferation and survival), and promote tumorigenesis [29].

A better understanding of membrane phospholipid metabolism in cancer cells will aid in understanding their role in cancer growth and identify if these changes could be manipulated for diagnostics and therapeutics [27]. Several studies have focused on identifying differences in phospholipid metabolism that develop during disease progression and the ability of phospholipid profiles to discriminate between normal and cancerous tissue [25,27,35,36,37]. Additionally, better characterization of altered phospholipid composition in cancer cells may serve as a useful diagnostic tool [38].

The aim of this review is to summarize the current research and literature regarding how phospholipid metabolism is altered in various human cancers, and to identify their potential as biomarkers and therapeutic targets for cancer. The focus of this review is on the two major bilayer forming phospholipids PC and PE. Dysregulation of PS in cancers is linked to its exposure on the surface of tumor cells and immunosuppression; this has been extensively studied and described in several recent reviews [39,40,41,42]. Phosphatidylinositol (PI) is also an important cellular signaling phospholipid, and dysregulation of PI in cancer has been also extensively studied and recently described [43,44,45].

## 2. Phospholipid Metabolism over the Course of Cancer Progression

### 2.1. PC Metabolism and CKα Are Elevated in Cancers

Elevated PC metabolism is an important hallmark of cancer [38,46,47]. Activated choline metabolism is associated with breast cancer malignancy and tumor progression, and differences in PC metabolite levels could be detected in early carcinogenesis [47,48]. Glycerol-phosphocholine is the major choline metabolite in normal cells, whereas in both immortalized and oncogene transformed cells there is a clear switch to phosphocholine as the dominant metabolite [47]. PC is generally increased in colorectal cancer, but PC(16:0/16:1) is specifically increased with more advanced disease stages [46]. 

Phosphocholine, which is a substrate for PC synthesis but also a product of PC metabolism by various phospholipases, is increased in advanced tumors and may serve as an indicator of tumor grade [37]. CKα, which produces phosphocholine from choline, is overexpressed in most cancers and increased CKα results in increased phosphocholine and cancer progression [48,49,50,51,52]. Overexpression of CKα causes a more aggressive cancer phenotype, increased invasion, and drug resistance in breast cancer cells [49,50]. In one study, colorectal cancer tissues showed higher levels of CKα than adjacent non-cancerous tissues derived from the same patient [51]. CKα expression was higher in advanced versus early-stage tumors and high levels of CKα correlated with tumor metastasis; however, it showed no correlation to tumor size, tumor grade, or local invasion. In both the early and advanced-stage patient groups, high CKα protein expression was associated with poor survival [51]. While this study suggests that CKα may predict poor patient prognosis, it is not currently considered a prognostic marker in colorectal cancer [52]. 

### 2.2. Cancer PE Metabolism Is Complexly Regulated at the Level of PCYT2

PCYT2 expression and activity are mostly reduced in cancers [53,54], but in breast adenocarcinoma cells metabolic stress leads to increased PCYT2 expression and activity [55]. Decreased PCYT2 activity results in decreased PE synthesis by the Kennedy pathway as well as accumulation of the substrate phosphoethanolamine [53,54]. Phosphoethanolamine stimulates cell growth and tumor progression [53], and phosphoethanolamine levels are established to be higher in breast cancer tissues [38]. A recent study has unveiled a novel role for PCYT2 as a glycerol-phosphate cytidylyltransferase [56]. This discovery is significant, as CDP-glycerol inhibits the glycosylation of α-dystroglycan [57]. When PCYT2 expression was reduced, CDP-glycerol was reduced, and therefore, the expression of the glycosylated form of α-dystroglycan was higher [56]. α-Dystroglycan is a laminin receptor frequently downregulated in breast cancer, and the loss of α-dystroglycan correlates with higher tumor stage and greater proliferation [57,58]. Therefore, this novel function of PCYT2 may be associated with tumor progression, in addition to its well-established role in PE synthesis by the Kennedy pathway, which will be discussed more in the next sections.

#### *PCYT2* Alternative Splicing Is Modified in Cancer Cells and Tissues

Increased splicing events correlate with cancer development, malignancy, and tumor progression [59,60]. A recent analysis of 150 triple-negative breast cancer patients identified seven alternatively spliced genes associated with overall survival [61]. One of these genes was the *PCYT2* gene, highlighting the importance of PCYT2 in relation to breast cancer [61].

PCYT2 is essential for survival, and its activity is regulated via post-translational modifications [62] and alternative splicing [55,62,63]. PCYT2 protein alone is not currently considered a prognostic marker for breast cancer, although this may not be accurate due to the nature of antibodies used to investigate this [64] and as will be shown here, the presence of multiple isoforms and the complexity of PCYT2 splicing.

Two functional PCYT2 proteins, α and β, are present in humans [62,63]. A third variant has recently been identified that produces a truncated PCYT2 protein (p.Arg377Ter) with reduced activity, and this mutation causes hereditary spastic paraplegia [65,66]. Three subsequent PCYT2 variants (Lys319Asn; Lys319Asn/Val320ins34; and Val303Ter) have been identified in patients with hereditary spastic paraplegia [67].

We characterized *PCYT2α* and *PCYT2β* expression and splicing in noncancerous mammary epithelial cells (MCF-10A), estrogen-dependent, non-invasive breast cancer cells (MCF-7), and triple-negative and aggressive breast cancer cells (MDA-MB-231) (Figure 2, Appendix A). The data shows that the splicing of the PCYT2 gene is altered in cancer cells, as evidenced by diminished *PCYT2α* mRNA and increased *PCYT2β* mRNA in breast cancer cells. (Figure 2A). PCYT2α protein was expressed only in control MCF-10A cells. Since PCYT2α and PCYT2β proteins are identical except for the linker peptide sequence, to which the PCYT2α-specific antibody binds [55], PCYT2β expression was deduced after probing with anti-PCYT2_total(α+β)_ antibody. Considering that there is little to no PCYT2α in the cancer cells, the lower size protein detected by anti-PCYT2_total(α+β)_ is PCYT2β, and it was highly expressed in MDA-MB231 cancer cells.

The mRNA product named *PCYT2γ* (Figure 2A) which was present only in normal cells was identified as *PCYT2-213* (Ensembl: ENST00000573401.6). Like *PCYT2α*, this variant contains exon 7 but also retains 221 bp from the end of intron 7 (Figure 3). *PCYT2γ* is classified as a nonsense-mediated decay biotype in the Ensembl database. Nonsense-mediated decay is a quality control pathway that degrades mRNAs that contain premature termination codons, but this function has yet to be confirmed for PCYT2γ [68,69,70].

Figure 3 shows the structure of additional *PCYT2* variants from the Ensembl database (ENSG00000185813). We investigated for the first time how are these transcript variants expressed in breast cancer cells. As shown in Figure 4 and Table 1, both protein-coding (*PCYT2-201*, *PCYT2-209*, *PCYT2-210*, *PCYT2-212*, *PCYT2-214*) and protein non-coding (*PCYT2-207*, *PCYT2-208*, *PCYT2-211*, *PCYT2-216*) variants were upregulated in cancer cells relative to control cells; *PCYT2-217* was not detected in any cell line. Since *PCYT2α*, *PCYT2-213*/*PCYT2γ*, and *PCYT2-205* were almost exclusively expressed in control cells (Figure 2 and Table 1), the data clearly demonstrated that deviations in the *PCYT2* splicing mechanism are an important aspect of breast cancer development.

## 3. Kennedy Pathway Enzymes as Prognostic Markers in Multiple Cancers

Phospholipid abnormalities are present in the early stages of disease, and alterations in the expression of key genes with roles in phospholipid synthesis can predict patient survival [71,72]. Increased expression and activity of CKα are associated with increased proliferation and malignancy and are an indicator of patient survival in NSCLC [73,74]. In early-stage NSCLC, patients with higher CKα had a greater risk of death compared to patients with lower CKα, who demonstrated improved survival [74].

Analysis of *PEMT* in lung cancer tissues relative to adjacent non-cancer lung tissues revealed that greater *PEMT* expression ratios were associated with shorter patient survival. The deceased patients were also found to have a 32% higher *PEMT* expression ratio compared to surviving patients [72].

Elevated *ETNK1* (ethanolamine kinase 1), is correlated with worse prognosis and survival among lung adenocarcinoma patients [75]. The CDP-ethanolamine Kennedy pathway genes *ETNK1*, *PCYT2*, and *SELENO1* are upregulated in both lung adenocarcinoma and squamous cell carcinoma tissues [76]. However, despite the observed differences in survival and disease prognosis these proteins are not deemed to be prognostic in lung cancer [52,77,78,79,80]. However, they are deemed clinically prognostic in other types of cancer, as we summarized in Table 2 [52,64,77,78,79,80].

## 4. Phospholipid Content and Composition Can Differentiate Cancer Subtypes

### 4.1. Phospholipid and Fatty Acid Profiling of Cancer Cells and Tissue Biopsies

An early study examining the lipid composition of breast tissue revealed that cancerous samples contained significantly more PC, PE, and sphingomyelin (SM), a choline-containing phospholipid, compared to a control group [31]. Furthermore, the fatty acid composition of the phospholipids was also altered. Cancer PC contained more short-chain fatty acids, and PE contained more unsaturated fatty acids [31]. Other research has confirmed that the fatty acid composition of phospholipids is significantly different in breast cancer tissues [32,81]. Patients with different types of breast cancer contained lower total levels of myristic acid, oleic acid, linoleic acid, and α-linoleic acid, while levels of stearic acid, n-3 polyunsaturated fatty acids, and n-6 polyunsaturated fatty acids were significantly higher in the cancer tissues [32]. Docosahexaenoic acid levels were high in both interface and tumor tissues. Looking at individual phospholipids, the fatty acid profiles were considerably different. Oleic and arachidonic acids were significantly higher in PC, PE, PI, and PS fractions from cancer tissues [32].

A recent study suggests that fatty acid remodeling of phospholipids may be an adaptive response to the acidic microenvironment observed in most tumors [81]. Several PC and PE species containing longer fatty acids were detected at higher levels in the breast cancer cells. The melanoma and prostate cancer cells also displayed altered levels of select PC and PE species [81]. Furthermore, fatty acid metabolism is heavily modified in cancer cells, including increased de novo fatty acid synthesis, enhanced fatty acid oxidation, and increased fatty acid uptake [82,83].

Patients with NSCLC have significantly reduced total PC, PS, and SM in cancer tissues [27]. The cancer tissues exhibited increased levels of specific PI, PE, and PC species and decreased levels of some SM species in the tumor tissues. Tissue-imaging confirmed that discriminatory PI species were increased, and SM species decreased in the tumor regions but not in the adjacent non-malignant regions [27]. Similarly, distinct phospholipid profiles for adenocarcinomas and squamous cell carcinomas could differentiate between the two subtypes [27]. A separate study confirmed that there are clear differences in key discriminatory lipids according to histological subtype [35]. One PC species, in particular, was noted to be highly overexpressed in adenocarcinomas, more than three times the levels observed in squamous cell carcinomas [35].

### 4.2. Phospholipid and Fatty Acid Profiling of Patient Plasma

In addition to variations in cancerous and normal tissues, there are also changes in phospholipid content in the bodily fluids of cancer patients [38]. For instance, plasma concentrations of PC and ether-linked PC are higher in patients with breast cancer [84]. Lysophosphatidylcholine (LPC) and cholesterol ester are lower in breast cancer patients compared to benign patients [84]. Low LPC in cancer patients is linked with increased inflammation and acts as an indicator of disease severity and poor clinical outcome [84].

Changes in plasma phospholipids and fatty acids also correlate with the development of colorectal cancer [36,85]. Higher content of the two monounsaturated fatty acids, elaidic acid, and palmitoleic acid, was associated with the formation of adenomas compared to patients who had no colon polyps [85]. In contrast, the content of palmitic acid, a saturated fatty acid, was lower in patients with adenomas [85,86]. Total fatty acids are significantly reduced in cancer patients, with decreased levels of saturated, monounsaturated, and polyunsaturated fatty acids. The plasma phospholipids also had lower levels of linoleic acid in cancer patients [86].

Exosomes are membrane-bound vesicles produced by budding and released into the extracellular environment [87]. Cancer patients have elevated exosome amounts in their blood [38,84]. Tumor-derived exosomes from breast cancer patients display altered phospholipid content compared to healthy individuals [38]. Given that exosomes are present in many bodily fluids (blood, urine, saliva), circulating plasma phospholipids from tumor-derived exosomes could serve as diagnostic biomarkers in blood-based screening [38].

Plasma metabolic profiling of breast cancer patients revealed that phospholipid metabolism was altered in four different breast cancer subtypes (luminal A, luminal B, triple-negative, and HER-2 positive). Among them, several PC and PE metabolites are differentially regulated in distinct breast cancer molecular subtypes [88]. In a recent study examining copy number alterations and gene expression profiles, *KIAA1967* and *MCPH1* were found to have a high correlation in all four breast cancer subtypes [89]. However, several subtype-specific genes were also identified. This included *PCYT2*, which was found to have a high correlation for the HER-2 positive breast cancer tumor subtype [89]. Evidently, both phospholipid and gene profiling represent an important tool for diagnostics of different breast cancer phenotypes.

While there is strong potential for fatty acids and phospholipid profiling to serve as diagnostic and prognostic markers for cancer development, further work is required to develop successful diagnostic tools. Yonekubo et al. (2010) sought to develop a lipid microarray protocol for the early diagnosis of breast cancer. Unfortunately, there were no differences identified in serum responses to lipids between wild-type and breast cancer samples [90]. In contrast, current advances in mass spectrometry-based technologies are the most promising methods for clinical diagnostics of lipids [91]. Such strategies are appealing as mass spectrometry-based lipidomic analyses are highly sensitive and capable of identifying changes in the levels and composition of several hundreds of various lipid species [92].

## 5. Kennedy Pathway Inhibitors as Novel Cancer Therapeutics

### 5.1. Targeting the Pathway Regulators

CKα has been extensively studied as a target for cancer therapies, and several CKα inhibitors with antitumor activity have been identified [48,93,94,95]. For example, the novel small molecule ICL-CCIC-0019 inhibits growth and reduces cell survival in several cancer cell lines, and suppresses tumor xenograft growth in mice [96]. The siRNA knockdown of CKα induces apoptosis in various cancer cells [50] and decreases the proliferation of triple-negative breast cancer cells [97]. In addition, in MDA-MB-231 tumor-bearing mice, the use of direct (H89) and indirect (sorafenib) CKα inhibitors, or anti-CKα shRNA decreased choline content, demonstrating the effectiveness of these treatments in vivo.

Choline transporter 1 (CTL1) has been also proposed as a molecular target for anti-cancer therapies [98,99]. CTL1 is overexpressed in several cancer cell lines and is associated with malignant progression [100]. A recent study reported that CTL1 also appears to have a dual function as an ethanolamine transporter [101]. The use of the choline analog hemicholinium-3 inhibits choline uptake and reduces cell proliferation in colon cancer cells [98]. Similarly, cell viability was reduced in NSCLC cells following hemicholinium-3 treatment [97]. Other CTL1 inhibitors continue to be explored, with many showing promising antitumor activities [98,100,102].

PCYT2 has been also proposed as a target for novel cancer therapies [103,104]. Meclizine, a direct inhibitor of PCYT2, is a potential anticancer drug when used in conjunction with the phosphofructokinase (PFKFB3) inhibitor PFK158 [103]. PFK158 inhibits PFKFB3, a key enzyme for glycolysis, reprogramming cellular metabolism to be more dependent on mitochondrial oxidative phosphorylation [103]. This renders cells more susceptible to meclizine-induced cytotoxicity, as inhibition of PCYT2 leads to an accumulation of phosphoethanolamine, which in turn disrupts the mitochondrial electron transport chain and thus, inhibits mitochondrial oxidative phosphorylation [103]. In a human hepatocarcinoma cell xenotransplantation model, the combination of PFK158 and meclizine reduced liver tumors [103].

Recently, we developed a new PCYT2 inhibitor CHY-1 as an antitumor drug candidate to treat lung cancer [104]. CHY-1 treatment had several effects on the lung cancer NSCLC cells, including reduced cell viability, cell cycle arrest, and induced ER stress [104]. Compared to other drugs (miltefosine, edelfosine, or ilmofosine), CHY-1 was more potent against cancer cells while also being less cytotoxic for noncancerous cells, as well as having a lower hemolytic activity. Using mouse models infected with LL/2 lung carcinoma cells, CHY-1 treatment proved effective in vivo, delaying tumor progression and growth [104]. CHY-1 inhibits PCYT2 activity causing reductions in de novo PE synthesis. PE is an essential ER, mitochondrial, and autophagosome bilayer phospholipid and a specific regulator of autophagy. CHY-1 reduces the autophagy, ER, and mitochondria-mediated processes in NSCLC cells. CHY-1 also could induce immunogenic cell death in NCSLC cells. To validate the target, we also deleted PCYT2 in lung cancer cells LL2/LC1 using CRISPR-Cas9 [105]. Wild-type and scrambled CRISPR-Cas9 cells developed tumors in nude Balb/c mice in 15 days. PCYT2 CRISPR-Cas9 knockout tumors showed delayed development in 28 days and continued to grow slower at 34 and 37 days. With this new type of antitumor activity, specific inhibitors of PCYT2 and the CDP-ethanolamine Kennedy pathway offer a promising novel treatment strategy for lung cancer [104].

### 5.2. Ethanolamine and Phosphoethanolamine

The PE precursors ethanolamine and phosphoethanolamine have been gaining interest for their antitumor activity [106]. Synthetic phosphoethanolamine (Pho-s) has been shown to be effective against several cancer cell lines, including Ehrlich ascites tumor [107], renal carcinoma [108], and breast cancer cells [109]. While the mechanism of action of Pho-s remains unclear, this agent has been found to have anti-proliferative activity, anti-metastatic, and does not exhibit cytotoxic effects on normal cells [107,108]. Pho-s inhibits tumor growth and prolongs survival in Ehrlich ascites tumor-bearing mice [107] and melanoma-bearing mice [110].

A non-toxic, orally deliverable product of ethanolamine has been developed that is effective in vivo [106]. After four weeks of ethanolamine treatment, prostate xenograft mice displayed a reduction in tumor size. Colon xenograft mice showed a decrease in tumor size after only two weeks of treatment. Following knockdown of CK, the ethanolamine treatment was far less effective, showing that the accumulation of phosphoethanolamine is responsible for the anti-tumor effects [106].

ETNK1 mutations have been found to occur in leukemia patients, diffuse B-cell lymphomas and in patients with systemic mastocytosis [111]. Mutated ETNK1 altered mitochondria morphology without a significant effect on phospholipids and fatty acid composition. Mutant ETNK1 induced mitochondria hyperactivation, ROS production, and DNA damage as a result of lower phosphoethanolamine production due to reduced ETNK1 enzyme activity. This was supported by the fact that supplementation with exogenous phosphoethanolamine was able to revert these effects [111].

### 5.3. Other Targets for Cancer Therapies

The synthetic fatty acid 2-hydroxyoleic acid (2OHOA) has shown promise for the treatment of glioma [112]. This molecule activates sphingomyelin synthase, restoring SM levels in cancer cells and decreasing PE content [112,113,114]. 2OHOA also alters the fatty acid profile in tumor cells and reduces oleic acid content in PC and PE [113]. 2OHOA induces autophagy by activating stress response pathways [115], as well as inhibiting several pathways responsible for cell proliferation, growth, and metabolism [114]. While 2OHOA displays specific and efficient activity against glioma cells, it has low toxicity for non-tumor cells [114,115], and clinical trials show promising results for 2OHOA as a therapeutic tool for glioma treatment [112].

Ophiobolin A (OPA) is a plant natural product that has been proposed as an anticancer treatment against glioblastomas [3]. The mechanism of action of OPA is to induce membrane leakiness and increase membrane permeabilization [116]. Furthermore, when the CDP:ethanolamine Kennedy pathway was inhibited through the inactivation of any of the three key enzymes (ETNK, PCYT2, EPT), myeloid cancer cells became resistant to OPA treatment [116].

Research continues to explore phospholipids and their metabolic pathways as potential avenues for anticancer therapies [98,117,118]. Various strategies targeting important enzymes such as ETNK, PEMT, PC-specific phospholipase D1, phospholipase C, sphingomyelinases, choline transporters, and glycerophosphodiesterases have been described [90]. Moreover, several peptides, (e.g., cinnamycin, duramycin, or cyclotides) have been identified that specifically bind PE in cell membranes, disrupting the membrane and eventually resulting in cell death [118].

### 5.4. Clinical Imaging

Since cancers have an active choline and PC metabolism, radiolabeled choline is widely utilized for the clinical detection and monitoring of tumors [119,120,121]. PET is a powerful imaging tool for the detection of tumors [122], and radiolabeled choline (^11^C-choline) in particular, is an important biomarker of tumors [120,123,124,125]. For instance, ^11^C-choline PET/CT has shown high sensitivity and specificity for the detection of lymph node involvement in prostate cancer patients [124,125]. Choline radiotracers also have a strong potential as prognostic markers. In a retrospective study of 210 prostate cancer patients, ^11^C-choline PET/CT was able to predict patients’ survival [120]. Follow-up with patients’ post-prostatectomy revealed that patients with a positive ^11^C-choline PET/CT had a median survival of 13.4 years while those with a negative ^11^C-choline PET/CT had very low fatal events.

^11^C-choline has higher sensitivity for HCC but given that ^11^C radionuclides have a relatively short half-life, choline derivatives labeled with ^18^F-fluorine, which have a longer half-life, can be used for HCC imaging. Several studies have found that ^18^F-fluoroethylcholine and ^18^F-fluorocholine show promising results in their ability to detect and stage HCC in cancer patients [123]. The use and effectiveness of ^18^F, ^11^C, and other PET radionuclides, along with various radiotracers for cancer imaging and detection have been reviewed in detail elsewhere [122].

Jaswal et al. (2022) recently developed a choline-based diagnostic probe for the clinical visualization of tumors. The probe, [^99m^Tc]Tc-DTPA-bis(ChoEA) is a synthetic choline analog that binds to the active site of CKα [119]. The intravenous administration of this conjugate did not have significant adverse health effects. The conjugate degraded slowly in serum and had high tumor uptake in the PC3 xenograft mouse model, showing great potential for clinical imaging of choline-rich tumors [119].

A fluorescent small-molecule choline mimetic JAS239 can also be used for optical imaging of tumors [126]. JAS239 binds and inhibits CKα activity and reduces cell growth comparable to MN58b, a well-characterized CKα inhibitor [126]. JAS239 proved effective at reducing tumor growth in vivo. Overall, JAS239 was able to determine CKα status in tumors, distinguish tumor margins, and evaluate MN58b efficacy, highlighting that fluorescent small-molecule diagnostics have multiple functions in the clinical setting.

## 6. Conclusions

The altered structure and function of cellular membranes reflect dysregulation of phospholipid metabolism in cancers. These changes have been associated with malignant transformations, tumor progression, and worse disease prognoses in a variety of cancers including breast, colorectal, and lung cancers. Of particular interest are the membrane-forming phospholipids, PC, and PE, as they show altered homeostasis in multiple cancers, and multiple genes from their metabolic pathways are dysregulated in cancers. Anticancer therapies targeting the key enzymes from the Kennedy pathway, such as CKα and PCYT2, show promise in various cancers. The changes observed in alternative splicing of PCYT2 in breast cancer cells highlight the importance of PCYT2 regulation. Given that not much is known about the PCYT2 splicing mechanism, more work is needed to fully characterize and understand the role of PCYT2 in cancer development.

Membrane phospholipid profiling can discriminate between normal and cancer tissues and to some degree can reflect the stage of cancerous transformation. Screening for differences in phospholipid species could be a potential biomarker in cancer diagnostics. Circulating phospholipids show potential for blood-based screening techniques, and lipidomic technologies as accurate and powerful analytic tools need to be more clinically utilized. Clinical research on this topic is however limited, and there are some conflicting data on how phospholipid metabolism is altered in certain types of cancer. Therefore, there is a need for further research into the regulation of phospholipid metabolism in cancers with the goal of potentially identifying genes and molecules that could serve as novel targets for cancer treatments.

## Figures and Tables

**Figure 1 ijms-23-05266-f001:**
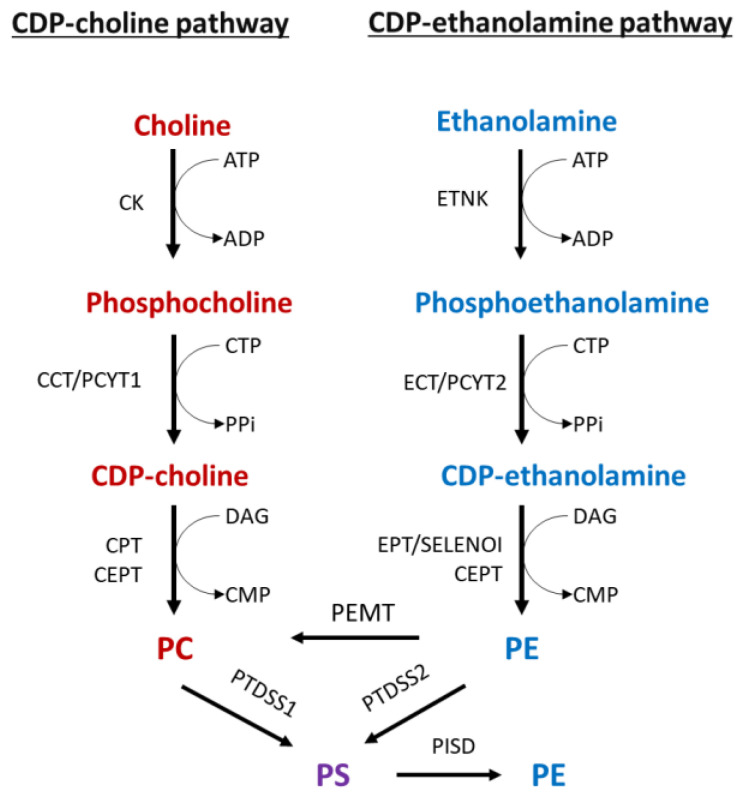
The formation of bilayer phospholipids by de novo pathways, methylation, and base-exchange reactions. Phosphorylation of choline by CK is the first step in the CDP:choline branchof the Kennedy pathway. Phosphocholine is then combined with CTP by CCT/PCYT1 to produce CDP:choline. This product is then coupled with DAG by CPT or CEPT to generate PC. PE is formed by the CDP:ethanolamine branch through analogous reactions that are catalyzed, sequentially, by ETNK, ECT/PCYT2, and EPT/SELENO1. PCYT1 and PCYT2 are the main regulatory enzymes of the Kennedy pathway. Additional PC can be produced by PE methylation using PEMT. PS is exclusively made from PC and PE by PTDSS1 and PTDSS2 base-exchange reactions. PS is decarboxylated to PE by PISD. Abbreviations: CK, choline kinase; CCT/PCYT1, phosphocholine cytidylyltransferase; CEPT, choline/ethanolamine phosphotransferase; CPT, choline phosphotransferase; DAG, diacylglycerol; ETNK, ethanolamine kinase; ECT/PCYT2, phosphoethanolamine cytidylyltransferase; EPT, ethanolamine phosphotransferase; PS, phosphatidylserine; PC, phosphatidylcholine; PE, phosphatidylethanolamine; PISD, phosphatidylserine decarboxylase; PS, phosphatidylserine; PEMT, PE *N*-methyltransferase; PTDSS1, phosphatidylserine synthase 1; PTDSS2, phosphatidylserine synthase 2; SELENO1 (selenoenzyme 1) ethanolamine phosphotransferase 1-EPT1.

**Figure 2 ijms-23-05266-f002:**
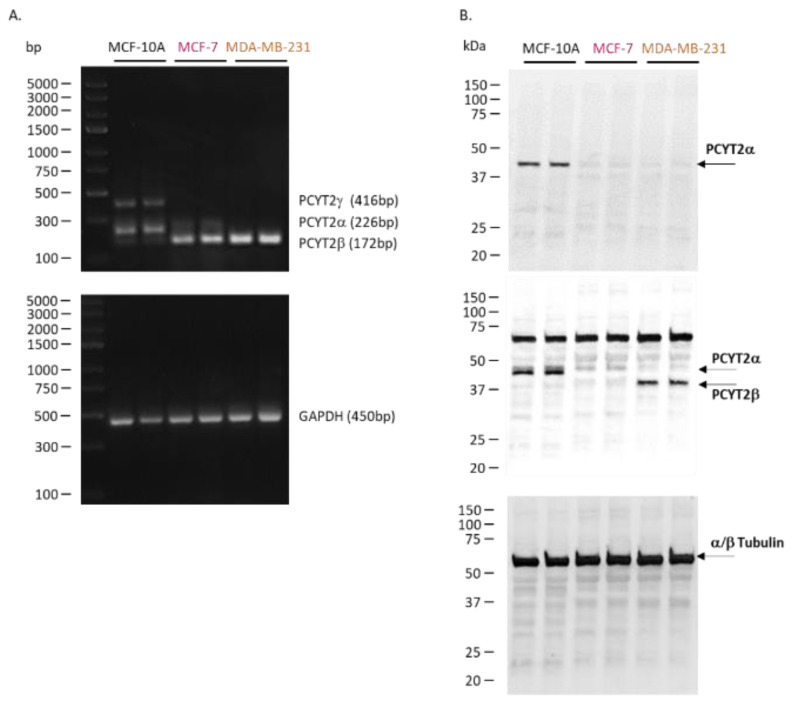
Catalytically active PCYT2 is not the same in normal and breast cancer cells. (**A**) *PCYT2α* is dominant in control MCF-10 cells while *PCYT2β* is dominant in breast cancer cells MCF-7 and MDA-MB231; *GAPDH* is a PCR control. (**B**) Immunoblotting with anti-PCYT2α, anti-PCYT2_total(α+β)_ antibodies confirm the high expression of PCYT2α and PCYT2β proteins in controls and MDA-MB231 cells, respectively; α/β tubulin is a protein control. A new transcript variant (*PCYT2γ*) was detected only in control cells.

**Figure 3 ijms-23-05266-f003:**
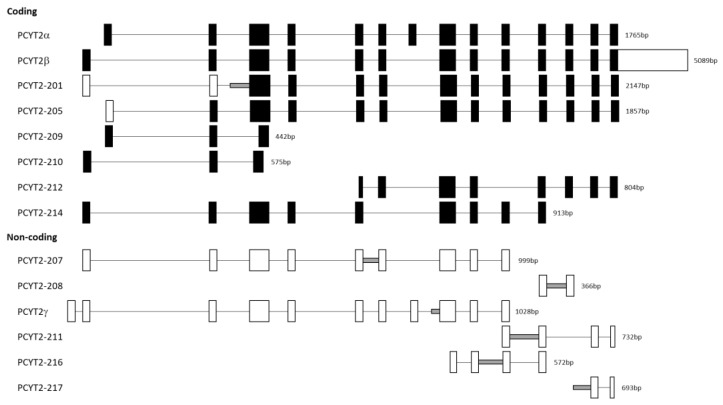
Exon-intron maps of *PCYT2* splice variants. The full-length *PCYT2* transcript (*PCYT2α*) has 14 exons. Coding and non-coding *PCYT2* variants (Ensembl databank: ENSG00000185813) are illustrated in comparison. Exons are shown by boxes, and introns are shown by straight lines. Retained introns/intron segments are represented by grey lines. The filled boxes represent coding regions, while empty boxes represent untranslated regions.

**Figure 4 ijms-23-05266-f004:**
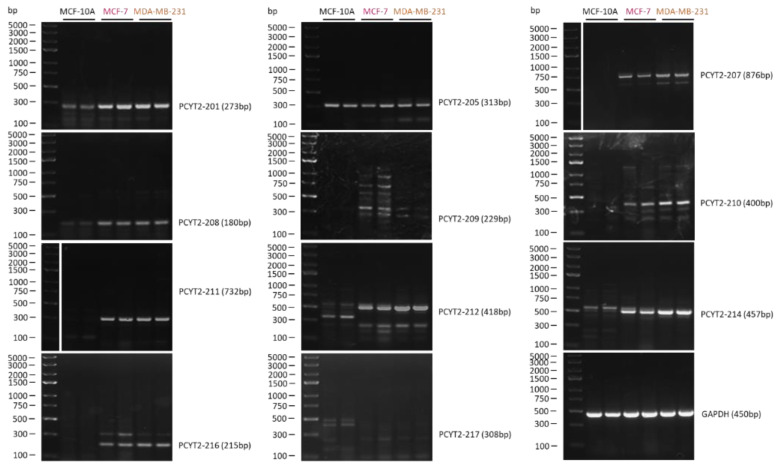
*PCYT2* splice variant expression in normal and breast cancer cells. MCF-10A control and MCF-7 and MDA-MB-231 breast cancer cells were grown in Dulbecco’s Modified Eagle Medium, high glucose, supplemented with 10% fetal bovine serum and 2% penicillin-streptomycin at 37 °C in a humidified, 95% air + 5% CO_2_ atmosphere. Total RNA was extracted from the cells and converted into cDNA. Amplification of *PCYT2* was done using variant-specific primers to show relative abundance of various *PCYT2* mRNAs.

**Table 1 ijms-23-05266-t001:** *PCYT2* splicing is altered in breast cancer cells. Splice variant number and biotype are from Ensembl databank: ENSG00000185813. Results are expressed as fold-change mRNA abundance relative to MCF-10A control cells.

**Upregulated Variants**		**Fold-Change Relative to Control**		
		**MCF-10A**	**MCF-7**	**MDA-MB-231**
Coding	*PCYT2-201*	1	8.73	8.32
Non-coding	*PCYT2β*	1	53 *	99 *
	*PCYT2-209*	0	2074 **	2925 **
	*PCYT2-210*	0	2840 *	2160 *
	*PCYT2-212*	1	39.5 *	68.0
	*PCYT2-214*	1	34.5 *	59.6 *
	*PCYT2-207*	0	1620 *	3380 *
	*PCYT2-208*	1	3.47 *	3.99 *
	*PCYT2-211*	1	36.5 **	51.8 *
	*PCYT2-216*	0	1889 **	2792 ** ^†^
**Downregulated variants**		**Fold-change relative to control**		
		**MCF-10A**	**MCF-7**	**MDA-MB-231**
Coding	*PCYT2α*	1	0.04 **	0 ** ^†^
	*PCYT2-205*	1	0.92	0.87
Non-coding	*PCYT2-213/* *PCYT2γ*	1	0 **	0 **

* *p* ≤ 0.05 compared to control; ** *p* ≤ 0.001 compared to control; ^†^
*p* ≤ 0.05 compared to MCF-7 as determined by two-tailed Student’s *t*-test.

**Table 2 ijms-23-05266-t002:** Cancer prognostic summary of the main phospholipid enzymes. Immunostaining was performed on cancer biopsies from 20 patients with *n* = 4–12 samples per cancer. Proteins whose expression is significantly (*p* < 0.001) associated with patient survival are determined to be a prognostic marker for that cancer. Data are obtained from The Human Protein Atlas (https://www.proteinatlas.org/ (accessed on 12 April 2021)).

Protein	Role	Cancer Type	Outcome	% of Patients with Increased Expression
**CKα**	Phosphorylation of choline	Liver cancer	Unfavorable	16
**ETNK1**	Phosphorylation of ethanolamine	Urothelial cancer Liver cancer	Favorable Unfavorable	25 16–33
**PCYT2**	Formation of CDP-ethanolamine	Renal cancer	Favorable	33–36
**PEMT**	Methylation of PE to PC	Endometrial cancer	Favorable	33
**SELENO1**	Formation of PE	No	N/A	N/A

## Data Availability

Additional data is available upon request.

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
