# Peer review of "Bilayer Forming Phospholipids as Targets for Cancer Therapy"

_ijms, 2022, doi:10.3390/ijms23095266_

Round 1
Reviewer 1 Report
This review investigates an interesting topic and could fill a gap in the literature by summarizing the researches on changes in phospholipid metabolism over oncogenic transformation and on the potential for phospholipids as biomarkers for diagnostics or as anticancer therapy targets. However, it is opinion of the reviewer that this paper cannot be accepted for publication at present form; to make it acceptable a very thorough and careful revision of the structure and wording is needed.
In particular, changes in phospholipid metabolism in breast, colonrectal and non-small cell lung cancer are described with little logical thread and with many trivial sentences making the paper often muddled, tricky and difficult to read and understand. Clinical evidence is scarce, and some conclusions are too speculative respect to studies reported.
Moreover, the paragraph 3.2. is very long, contains too many detailed for a review and is not very conclusive. The paragraph 6.2 is also very long and covers a lot of ground, sometimes difficult to follow Contrarily, the paragraph 5.1 is too superficial and does not provide any metabolic information.
In addition, there are several errors in the legend of fig. 1 and there are some typing errors in text.
Author Response
1. In particular, changes in phospholipid metabolism in breast, colorectal, and non-small cell lung cancer are described with little logical thread and with many trivial sentences making the paper often muddled, tricky, and difficult to read and understand. Clinical evidence is scarce, and some conclusions are too speculative with respect to studies reported. Thank you for your valuable comments. We have practically reworked the entire text to get our point across more clearly and to better connect ideas when discussing the relationship between membrane lipid regulation in cancers. Clinical evidence had already been included throughout the text in the utilization of lipid profiling of patient plasma and tissue biopsies, correlations between patient tumor size/tumor grade and lipid enzyme expression, and showing what tumor proteins are prognostic/diagnostic in human cancers, but this has now been better emphasized. We have also included a section dedicated to discussing advances in drug development and clinical imaging focusing on choline radiotracers and probes. 2. The paragraph 3.2. is very long, contains too many details for a review, and is not very conclusive. Paragraph 6.2 is also very long and covers a lot of ground, sometimes difficult to follow. Contrarily, paragraph 5.1 is too superficial and does not provide any metabolic information. Several sections of the text including the paragraphs of concern have been edited and condensed. The paper overall has been restructured for clarity. 3. In addition, there are several errors in the legend of fig. 1 and there are some typing errors in the text. Typing errors have been fixed.

Reviewer 2 Report
Remarks about the manuscript:
The authors have presented an excellent review article " Phospholipid metabolism in cancer cells and their potential as targets for cancer therapies" This review highlighted recent studies in alternate cancer treatment targeting membrane lipids. The Membrane lipids-based treatment is a relatively new exploring field, and this review included proper references to highlight their importance. Most of the references in this review are focused on PC and PE with other lipids regulation in membrane and their effect on the cancer microenvironment. The data submitted in this manuscript is superb, with a proper explanation through appropriate references and futures prospect.
Although this manuscript contains all the information to Publish in the IJMS journal, I have some questions:
- Information about negatively charged membrane lipids (PI, PS) is limited while they play a crucial role in many cellular mechanisms related to cancers. It would be much more informative if the author included some references to these lipid's functions.
- The quality of Gel is not good in Figure 4 for PCYT2-209 and PCYT2-210 (especially the MCF-10A band).
Some technical improvements needed:
Short form of names and spacing between sentences need to cross-check.
1.Nomenclature of “PS” is the same for both Phosphatidic acid and Phosphatidylserine (page 2, lines 63 and 64), and two short forms (PISD and PSD) used for phosphatidylserine decarboxylase (page 2, line 64 and 66),
2.Please check the double-spacing at many places in the manuscript (Page 157, line 713)
This manuscript (ID: IJMS-1668931) can be accepted in its current form with an explanation of the above comments.
Author Response
1. Information about negatively charged membrane lipids (PI, PS) is limited while they play a crucial role in many cellular mechanisms related to cancers. It would be much more informative if the author included some references to these lipid's functions.
We agree. We have clearly stated the focus of the review in the introduction, and provided references for other papers that have already reviewed PI and PS. Our focus was on PC and PE.
2. The quality of Gel is not good in Figure 4 for PCYT2-209 and PCYT2-210 (especially the MCF-10A band).
Figure 4 has been updated. There are no bands for PCYT2-209 and PCYT2-210 in MCF-10A cells.
3. Short form of names and spacing between sentences need to cross-check.
Abbreviations have been corrected in both captions and text. We followed the journal requirements and the spacing in the manuscript has been verified and edited to match desired format.
4. Nomenclature of “PS” is the same for both Phosphatidic acid and Phosphatidylserine (page 2, lines 63 and 64), and two short forms (PISD and PSD) used for phosphatidylserine decarboxylase (page 2, line 64 and 66),
Please see the above comment, this has been fixed.
5. Please check the double-spacing at many places in the manuscript (Page 157, line 713)
The spacing in the manuscript has been verified and edited to match desired format.

Round 2
Reviewer 1 Report
The manuscript submitted by Stoica et al. titled "Bilayers Forming Phospholipids as Targets for Cancer Therapy" has been greatly improved since the last submission. Now this review is well-written and changes in phospholipid metabolism over oncogenic transformation and the potential of phospholipids for cancer diagnosis or therapy are more clearly appreciable by the readears.
In my opinion the paper, in this form, is now suitable for publication.